# Effect of Lactylated Gluten and Freeze-Thaw Cycles on Frozen Dough: From Water State and Microstructure

**DOI:** 10.3390/foods12193607

**Published:** 2023-09-28

**Authors:** Yan Li, Yu Wang, Xi Qiu, Mingcong Fan, Li Wang, Haifeng Qian

**Affiliations:** State Key Laboratory of Food Science and Technology, School of Food Science and Technology, National Engineering Research Center for Functional Food, Jiangnan University, 1800 Lihu Avenue, Wuxi 214122, China; liyan0520@jiangnan.edu.cn (Y.L.); 6200112081@stu.jingnan.edu.cn (Y.W.); 18606200595@163.com (X.Q.); 8202009403@jiangnan.edu.cn (M.F.); wangli@jiangnan.edu.cn (L.W.)

**Keywords:** lactylated gluten, freeze-thaw tolerance, steamed bread

## Abstract

The influence of lactylated gluten and Freeze-Thaw Cycles on the water state, microstructure, and quality of frozen steamed bread dough was investigated. After three freeze-thaw cycles (3F/T), the specific volume of steamed bread with sodium lactate-treated gluten increased by 18.34% compared with the blank group and 5.73% compared with the wheat gluten (WG) group. Compared with wheat gluten, the texture properties of steamed bread with lactylated gluten increased significantly. Changes in rheological properties demonstrated that the frozen dough’s viscoelasticity increased significantly. The lactylated gluten could reduce water mobility and decrease the content of freezable water in frozen dough. Moreover, the free sulfhydryl (SH) content increased, revealing that the protein was depolymerized. Based on the microstructure and corresponding protein network analysis (PNA), the total area and the number of protein network connection points of the dough adding lactylated gluten were significantly higher than those of the blank group and the WG group. In conclusion, lactylated gluten enhanced the freeze-thaw tolerance of frozen dough.

## 1. Introduction

Frozen dough refers to a technology using a quick-freezing method in a centralized facility to produce frozen dough on a large scale. Then, it is transported to local retail stores or chain markets to complete the remaining processing process. In recent years, the global food market has witnessed a growing demand for frozen dough products, thanks to the advancements in frozen processing technology and the continuous enhancements in cold chain transportation [1]. Although frozen dough has many advantages, compared to fresh dough, the final product made from frozen dough will experience problems such as reduced specific volume, increased hardness, uneven pore structure, and decreased overall product acceptance. It is inevitable that there will be temperature fluctuations in the cold chain [1,2,3].

The decline in the quality of frozen dough products is mainly caused by ice crystals formed during freezing and ice crystal recrystallization resulting from temperature fluctuations [4]. In the process of repeated thaw and freeze, ice crystal molecules gradually grow larger, resulting in the destruction of the continuous and uniform gluten network structure. Meanwhile, the puncturing of yeast cells and the enlargement of the starch particle void result in a decline in the quality of the final product. A large number of studies have shown that the deterioration of the quality of frozen dough products caused by temperature fluctuations can ultimately be attributed to the change in dough components: gluten, yeast, starch, and other components. Among them, the destruction of the gluten network structure caused by freezing is the direct cause of dough quality deterioration [3,5,6].

The gluten network structure is important for the processing properties of dough and the quality of the final product. On the one hand, the relatively strong three-dimensional network structure can improve the gas-holding capacity and make the dough wrap more CO_2_. On the other hand, the tight gluten network can resist mechanical damage caused by ice crystals during the freeze–thaw cycle, making the final product have better texture characteristics. Xin et al. [7] found that the ice crystals formed during freezing destroyed the three-dimensional structure of the gluten network, resulting in the separation of starch particles and gluten proteins. Wang [8] found that freezing depolymerized glutenin macropolymers (GMP) resulted in a decrease in the viscoelasticity of gluten. The study of Frauenlob et al. [9] showed that the change in water state and the migration of water during freezing processing would damage the gluten network structure and impair the gas holding capacity of frozen dough, thus leading to the decline in the quality of the final product. Wang et al. [10] found that during the freezing process, the relatively ordered α-helix structure in the secondary structure of gluten was significantly reduced.

Strengthening the gluten network and making up for the damage to the network structure is an important means to protect frozen dough. As a natural reinforcer, wheat gluten protein can be used as an excellent improver for dough quality. After the addition of gluten, the strength of the dough is not only improved but the network structure is also more regular. The quality of frozen dough and the final product is significantly related to the integrity and stability of the gluten network structure. Wang et al. [11] found that the addition of GMP significantly improved the elasticity and gas retention, and the prepared steamed bread had higher specific volume and better texture attributes. Frauenlob et al. [9] studied the effects of eight different wheat flours on the quality of frozen bread. The results indicated that flours with a strong gluten network showed high ductility resistance and were most suitable for the production of frozen dough.

In summary, a large number of studies have shown that a strong gluten network structure is helpful in maintaining the quality of frozen dough and making the final steamed bread have a larger specific volume. In previous studies, we found that the weight loss of lactylated gluten was significantly lower than that of unmodified gluten. This might be because the connections of lactate groups caused gluten proteins to form stronger and tighter networks [12]. After lactylation, the viscoelasticity of gluten protein was also significantly improved [12]. Specifically, lactylation affected the change of protein–protein interaction and promoted the sulfhydryl disulfide bond’s exchange of gluten, forming a more extensive covalent crosslinking, and thus forming a more robust and tight network structure [12]. Therefore, the aim of this study was to apply lactylated gluten to slow down frozen dough’s deterioration.

## 2. Materials and Methods

### 2.1. Materials and Reagents

The commercial gluten was provided by Yihai Kerry (Jiangsu, China). The lactylated wheat gluten was prepared according to Wang et al. [12]. The yeast (Saccharomyces cerevisiae) was obtained from the local supermarket. All other reagents were of analytical grade.

### 2.2. Formulation of Dough and Preparation of Steamed Bread

The production formula of frozen dough consisted of 100 g flour, 50% water, and 1% yeast. The experimental dough needed the addition of 0.5% wheat gluten. First, the dry material was mixed in the flour mixer. Then, the yeast that was dispersed in water in advance was mixed with the dry material, and the flour was stirred for 7 min to obtain a smooth dough. The dough was divided into 50 g pieces and was kneaded round in shape. The dough was frozen in a −40 °C refrigerator for 3 h until the center temperature reached −20 °C, and then immediately transferred to a −18 °C refrigerator (Haier electric freezer BC/BD-519HAN, Qingdao, Shandong, China). A freeze–thaw cycle consists of freezing at −18 °C for 23 h and thawing at 30 °C for 1 h. During the experiment, 1~3 freeze–thaw cycles (1F/T~3F/T) were required. The fermentation room was used to adjust the temperature while thawing. In addition, part of the dough was dried in a vacuum freeze dryer (LGJ-10E, Beijing Sihuan Scientific Instrument Factory Co., Ltd., Beijing, China), crushed, and stored in a Ziplock bag for later use.

The steaming process of steamed bread: after the freeze–thaw cycle, the dough was fermented in a fermentation room. The fermentation condition was set to 37 °C and a relative humidity 80% for 1 h. The fermented dough was steamed in a steamer for 20 min to obtain the final steamed bread. After the fresh dough was shaped, it went directly into the fermentation and steaming process (Medei SYH28-21, Foshan, Guangdong, China).

The samples were abbreviated as a blank group: WG group (wheat gluten was added), Lac-WG group (wheat gluten treated with lactate was added), and NaLa-WG group (wheat gluten treated with sodium lactate was added).

### 2.3. Determination of Steamed Bread’s Properties

#### 2.3.1. Measurement of Specific Volume

The steamed bread was allowed to cool for 1 h at room temperature before its mass was measured. Additionally, the millet replacement method was used to determine the volume of the steamed bread [13]. The specific calculation formula is as follows:$$\mathrm{SV}(mL/g)=\frac{V}{m}$$
where SV was the specific volume, mL/g; V was the volume of steamed bread, mL; and m was the mass of steamed bread, g.

#### 2.3.2. Measurement of Texture Properties

After cooling the steamed bread for one hour, the texture properties of the bread were evaluated using a texture analyzer (BOSINTECH, Shanghai, China). The texture analyzer used for the measurements was slightly adjusted based on the procedure described by Wang [14]. The texture profile analysis (TPA) mode was conducted at a pre-test speed of 2.00 mm/s, mid-test speed of 1.00 mm/s, and post-test speed of 1.00 mm/s, with a trigger force of 5 g. The steamed bread was cut into uniform slices with a thickness of 15 mm by a microtome. The specific parameters of the physical property analyzer were set as follows: cylindrical probe P36 and distance strain 25%.

#### 2.3.3. Measurement of Color

After the steamed bread was cooled for 1 h, the apparent color was measured with a color difference meter (CR8+, Shenzhen 3nh Technology Co., Ltd., Shenzhen, China). It was calibrated before measuring by the white and black standard tiles. The determined indicators were L* (0: black; 100: white), a* (−a*: greenness; +a*: redness), and b* (−b*: blueness; +b*: yellowness). The measurements of every sample were performed in three replicates.

### 2.4. Measurement of Rheological Property of Frozen Dough

Part of the samples were placed on the dynamic rheometer platform with the measurement gap set at 2000 μm. After lowering the 40 mm flat fixture to a height of 2050 μm, excess samples were removed and silicone oil was applied on the edge to prevent water loss. A 1% strain was selected in the linear viscoelastic region, and the frequency range was 0.1~10 Hz. The balance time of gluten protein was 300 s, and the test temperature was 25 °C.

### 2.5. Measurement of the Water Distribution of Frozen Dough

In the study conducted by Xin et al. [7], the water distribution of frozen dough was analyzed using a low-field nuclear magnetic resonance (LF-NMR) technique. To perform the analysis, 1.5 g of the dough sample was carefully wrapped in polytetrafluoroethylene sealing tape and subsequently placed inside a nuclear magnetic resonance tube. The FID pulse sequence was used for calibration, and the CPMG pulse sequence was used for determination. The specific parameters were set as follows: TE = 0.25 ms, TW = 1500 ms, TD = 75,032, NECH = 1500, and NS = 2. After the collection was completed, the T_2_ relaxation time curve was obtained by inversion of the obtained data, and the iteration times were 1 million times.

### 2.6. Measurement of the Freezable Water Content of Frozen Dough

The determination of freezable water content in dough was conducted using a modified version of the method reported by Ding et al. [15] and utilizing a differential scanning calorimeter. First, the center of the dough was put into the crucible and sealed, and then the freeze–thaw cycle was carried out according to the above conditions. The DSC was programmed as follows: the system routine was held at −20 °C for 3 min and heated from −20 °C to 10 °C at a rate of 2 °C/min to obtain the peak enthalpy value (ΔH) of the endothermic curve. The freezable water ratio (FW) was calculated using the following formula:$$F_{W}\left(\%\right)=\frac{∆H}{{∆H}_{w}\;\times \;W_{t}}$$
where ΔH_w_ was 333 J/g and W_t_ (%) was the moisture content of different doughs.

### 2.7. Measurement of the Free SH Content of Frozen Dough

According to Wang et al. [12], first, the Tris-glycine-EDTA buffer was prepared. Then, 10.4 g Tris, 6.9 g glycine, and 1.2 g EDTA were dissolved in deionized water and the volume was fixed to 1000 mL. Then, the pH was adjusted to 8.0. Ellman reagent was prepared with 40 mg DTNB dissolved in 10 mL Tris-glycine-EDTA buffer. In addition, 8 mol/L urea solution and 8 mol/L urea–5 mol/L guanidine hydrochloride solution were prepared with Tris-glycine-EDTA buffer.

First, 75 mg sample was mixed with 1 mL Tris-glycine-EDTA buffer. Then, 4.7 g guanidine hydrochloride was added, and the buffer solution was fixed to 10 mL. Then, 1 mL of the above solution was added to 4 mL of urea-guanidine hydrochloride solution and 0.05 mL of Ellman reagent. The absorbance was measured at 412 nm after 30 min light shielding reaction at room temperature.
$$\mathrm{SH}\left(\mu M/g\right)=\frac{73.53\;A_{412}\times D}{C}$$
where A_412_ was the absorbance at 412 nm, D was the dilution factor (5.02), and C was the sample concentration (mg/mL).

### 2.8. Microstructure of the Gluten Network in Frozen Dough

Microstructure analysis was determined using a confocal laser scanning microscope (CLSM) (LSM 710, Leica Co., Wetzlar, Germany) according to the method of Lu et al. [4]. The central part of the frozen dough sample was fixed with gel and cut into 20μm slices. After staining with 0.1 mg/mL Rhodamine B solution for 30 s, the microstructure was observed by CLSM.

Protein network analysis (PNA) was performed on the acquired images using Angio Tool software (version 64, National Cancer Institute, Bethesda, MD, USA). Protein diameter parameters were set to 3 and 5, intensity threshold parameters were set to 15 and 255, and the “fill holes” function was turned off. The calibration was set to 4.76 pixels/µm [16].

### 2.9. Statistical Analysis

SPSS software (version 26, IBM Co., Armonk, NY, USA) was used for the significance analysis of the data, and *p* < 0.05 was considered to be statistically significant. Origin 2018 software was used for drawing.

## 3. Results and Discussion

### 3.1. Quality Characteristics of Steamed Bread

A large specific volume, soft texture, and uniform internal structure are ideal quality characteristics of steamed bread, among which specific volume is the most intuitive index to reflect the quality of steamed bread [17]. Steamed bread with different treatments is analyzed in Figure 1 for SV changes during freeze-thaw cycles. Steamed bread had a specific volume below the acceptable threshold (2.2 mL/g) after three freeze–thaw cycles. Compared with the relatively constant experimental conditions in the laboratory, the temperature fluctuation range of frozen dough during the actual transportation is more uncontrollable. The repeated freeze–thaw is more likely to lead to the decline of the specific volume of frozen raw steamed bread [18]. Therefore, in actual production, quality deterioration may be more serious [19]. In the experiment, with the increase of freeze–thaw cycles, the specific volume of all samples decreased significantly (*p* < 0.05). The blank group decreased from 2.83 mL/g to 2.18 mL/g, and the steamed bread with lactate-modified protein decreased from 2.89 mL/g to 2.58 mL/g. The repeated freezing and thawing of water molecules in the process of temperature fluctuations caused the ice crystals to grow larger. The mechanical action of the ice crystals punctured the yeast cells, reducing gas deliverability. At the same time, the enlarged ice crystals also caused strong damage to the gluten network and reduced the gas-holding capacity, resulting in the deterioration of the quality of the final product [20].

However, in every experiment, the SV of steamed bread with lactylated gluten was significantly higher than that of the blank group and WG group (*p* < 0.05), indicating that the freeze-resistance of frozen dough and the quality of steamed bread were improved. Specifically, after three freeze–thaw cycles, the specific volume of the blank group was 2.18 mL/g, and the specific volume of the dough with sodium lactate-treated gluten increased by 18.34%. The hydrophilic lactate group was introduced, which leads to the inhibition of the formation of recrystallization and thus reduces the destruction of gluten. At the same time, the combination of lactate groups and water molecules could also slow down the trend of increasing free water in the dough. Meanwhile, water and gluten were closely combined, and the dough’s freeze–thaw tolerance was enhanced. Some studies have shown that fish skin antifreeze polypeptide could also inhibit the formation of recrystallization, slow down the weakening of the gluten network, and thus make the frozen dough have a high specific volume. This is because the hydrophilic group of the amino acid side chain can bind with water molecules [20].

The freeze-thaw cycles are illustrated in Figure 2, displaying the variation in the steamed bread. Figure 2 shows the change in the hardness of steamed bread during the freeze-thaw cycles. As can be seen from the figure, compared with fresh steamed bread, the hardness of all products increased significantly with the increase in the number of freeze-thaw cycles (*p* < 0.05). The hardness of steamed bread was higher than the acceptable threshold after one freeze–thaw cycle, indicating that the freeze-thaw cycle made the steamed bread form a more dense and compact structure, and the texture quality deteriorated. This might be related to decreased yeast activity and degradation of the gluten network [21]. Compared with the WG group, after three freeze-thaw cycles, the hardness of the dough supplemented with lactylated gluten protein was significantly lower than that of the WG group (*p* < 0.05). Specifically, after three freeze-thaw cycles, the dough hardness of the blank group was 1476.22 g, and the dough hardness of the added WG, Lac-WG, and NaLa-WG was reduced to 1402.48 g, 1332.54 g, and 1272.39 g, respectively. This might be due to the improvement of viscoelasticity by adding lactylated gluten, which was related to the rheological properties of frozen dough.

An elastic product can return to its original shape after deformation under a certain pressure, and it is positively correlated with softness [22]. Table 1 shows the change in the elasticity of steamed bread during freeze–thaw cycles. The elasticity of steamed bread decreased significantly after freeze–thaw cycles (*p* < 0.05). The elasticity of the blank group decreased by 8.51% after three freeze–thaw cycles, indicating that steamed bread formed a more rigid structure after freeze–thaw cycles. The changing trend of elasticity of steamed bread might be related to the change in its specific volume, and the addition of lactylated gluten could significantly improve the decrease of elasticity caused by freeze–thaw cycles. Specifically, after three freeze–thaw cycles, the dough elasticity with the addition of NaLa-WG increased by 4.65% compared with the blank, and the dough elasticity with the addition of Lac-WG increased by 3.49% compared with the blank. This might be due to the high foaming and emulsifying ability of the lactylated gluten, which helped to maintain the good gas retention of the frozen dough.

The color of steamed bread is one of the important qualities of its appearance, which can be measured by a color difference meter. In the determination parameters of the color difference meter, L* represents the brightness, a* represents the red–green degree, and b* represents the yellow–blue degree. Unlike the dark brown surface characteristics of bakery products such as bread, the bright white apparent color of steamed bread is expected by consumers. As shown in Table 2, fresh steamed bread had the highest brightness value (85.82). As the number of freeze–thaw cycles increased, the brightness values decreased significantly (*p* < 0.05), indicating that the freeze–thaw cycles would make the steamed bread appear darker. Giannou et al. [23] pointed out that the change in the apparent color of steamed bread was related to enzymatic browning of the dough surface during freezing storage and freeze–thaw cycles. After three freeze–thaw cycles, the apparent color of steamed bread with lactylated gluten was relatively the best, which indicated that the lactylated gluten could resist some brightness reduction caused by freeze–thaw cycles.

### 3.2. Dynamic Rheological Properties

The change in dynamic rheological properties of frozen dough during the freeze–thaw cycles is illustrated in Figure 3. The storage modulus and loss modulus of the dough exhibited an increment with increasing frequency, with the storage modulus showing a greater increase compared to the loss modulus. This indicates that dough is a material with both elasticity and viscosity, and its elastic characteristics are stronger than the viscosity characteristics. The storage modulus and loss modulus exhibited a declining tendency as the number of freeze–thaw cycles augmented, aligning with the findings of Wang et al. [11]. In this study, the decrease of G′ and G″ in dough might be due to the temperature gradient inside frozen dough caused by repeated temperature changes, which caused water molecules to freeze and thaw repeatedly. In addition, the volume and quantity of ice crystal molecules increased, which promoted the polymerization of gluten molecules in dough and thus reduced its viscoelasticity [24]. As shown in Figure 3, during the freeze–thaw cycles, lactylated gluten could improve the viscosity and elasticity of dough, making it exhibit greater solid-like properties. This might be because the addition of lactylated gluten introduced hydrophilic lactate groups that could enhance the interaction between water and solids in the dough. Similarly, it could inhibit the growth and recrystallization of ice crystals and protect the rheological properties of the dough to form a more stable network structure. At the same time, the addition of lactylated gluten could enhance the degree of crosslinking of the gluten network in frozen dough, which was related to the change in disulfide bond content.

### 3.3. Moisture Distribution

During the process of freeze–thaw cycles, the transformation of water in the dough is primarily due to the creation and restructuring of ice crystals. This phenomenon significantly influences the internal conditions of the dough system and subsequently degrades the gluten content. Therefore, it is particularly important to understand the variation law of the water state for exploring the quality deterioration law of steamed bread. The distribution and migration of moisture in dough during freeze–thaw cycles can be studied by using LF-NMR to determine 1H relaxation characteristics. The three peaks on the LF-NMR relaxation curve represent the three states of water: T_21_ is strongly bound water, T_22_ is weakly bound water, and T_23_ is free water. The values of T_21_, T_22_, and T_23_ respectively indicate the relaxation time of water in different states in the dough. The smaller the value the lower the fluidity of the corresponding type of water, that is, the more closely the water is bound to the dough components. A_21_, A_22_, and A_23_ represent the proportion of T_21_, T_22_, and T_23_ moisture states to the total moisture of the dough, respectively [25].

Figure 4 shows the hydrogen proton group curve measured by LF-NMR of frozen raw dough with different gluten added during the same freeze–thaw cycle. Studies have shown that T_2_′ s shift towards a shorter relaxation time represented an enhanced interaction between the water molecules and the solid [25]. Figure 4A shows the T_2_ shift to the left of the frozen dough with lactylated gluten, indicating that the addition of lactylated gluten enhanced the interaction of water molecules with gluten due to the introduction of hydrophilic lacate molecules. Figure 4B–D shows the change of A_21_, A_22_, and A_23_ in the frozen dough. T_21_ represents the bound water closely related to the gluten matrix; T_22_ is the weakly bound water, filling the parallel space in the gluten; and T_23_ is the free water with high mobility, which is distributed in the gluten network voids [26,27]. As shown in Figure 4B, A_21_ of all samples showed a downward trend with the increase of freeze–thaw cycles. This might be due to temperature fluctuations that repeatedly thaw and freeze water molecules in dough, resulting in a decrease in the proportion of strongly bound water. However, in the same freeze-thaw cycle, lactylated gluten-fortified frozen dough’s A_21_ was significantly higher than that in the blank group and the WG group. This may be related to the ability of lactylated gluten to improve the degree of crosslinking of the gluten network; in this case, the gluten network’s interaction with water was enhanced. It can be seen from Figure 4C,D that with the increase in freeze-thaw cycles, the proportion of the semi-bound water and free water in the dough increased. And the results correspond to the change of A_21_, which indicated that the freeze-thaw cycle made the water state in dough tend to develop toward free water. However, the addition of lactylated gluten could inhibit water migration in frozen dough.

### 3.4. Freezable Water

The change of the water state in the dough affects the content of frozen water and then affects the final quality of steamed bread [28]. Figure 5 shows the influence of the different gluten additions on the freezable water content of frozen dough. In Table 3, ΔH is the enthalpy of the melting peak of the sample heat absorption curve, Wt is the water content, and Fw is the freezable water content. With the increase in freeze–thaw cycles, the content of the freezable water in the dough increased significantly. As can be seen from Table 3, there were two reasons for the increase in the content of freezable water. On the one hand, the increase in the number of freezing–thawing cycles caused the water molecules to freeze and thaw repeatedly. It resulted in an increase in the volume of ice crystals and the destruction of the gluten network structure, thus reducing its water content. On the other hand, repeated freeze–thaw cycles gradually increased the number of ice crystals, and the sample absorbed more heat when thawing the ice crystals, thereby increasing its ΔH. They combined to increase the final freezable water content. Kontogiorgos et al. [29] considered that freezing storage would induce protein dehydration and free water would eventually increase the freezable water content. Freezable water is formed by free water and a part of weakly bound water in dough. The volume of freezable water expanded during freezing, causing damage to the gluten network structure of the dough. The other part of weakly bound water and strongly bound water constituted non-freezable water, which would not freeze at a certain temperature, so it would not affect the structure of the gluten network. The freeze–thaw cycles led to an increase in freezable water content, which was consistent with the above moisture distribution results.

After the addition of lactylated gluten, the content of freezable water in the dough decreased significantly, indicating that the addition of lactylated gluten can effectively protect the frozen dough. After three freeze–thaw cycles, the content of freezable water in the Lac and NaLa group decreased by 11.02% and 9.03%, respectively, compared with the blank group, which might be related to the hydrophilicity of lactate groups. The hydrophilic lactate group could bind to water molecules, thus reducing the fluidity of water in dough. Less strongly bound water could be converted to free water, thus reducing the content of freezable water in dough [15]. This has a certain protective effect on the frozen dough and thus obtains steamed bread with larger volume and less hardness. SaAFP reduced the freezable water content of frozen dough by 12% because SaAFP’s hydrophilic groups bind to water [20].

### 3.5. Free SH

The degree of crosslinking of free SH in frozen raw dough plays an important role in maintaining the network structure of dough and has a significant impact on the final quality of steamed bread. Therefore, it is necessary to analyze the change in free SH content in dough [30]. Figure 6 shows the effect of the addition of lactylated gluten on the free SH of frozen dough after freeze–thaw cycles. During the whole freeze–thaw cycle, the free SH of the dough in the blank group increased from 2.99 μmol/g to 4.66 μmol/g (*p* < 0.05). Moreover, the free SH content in the other groups also showed a significant increase trend with the increase in the number of freeze–thaw cycles. It indicated that the disulfide bond was broken during the freeze–thaw process, and more free SH was exposed. Water redistribution and ice recrystallization weakened the disulfide bond [31]. For wheat flour dough, the increase of free sulfhydryl groups is not conducive to the quality of the final product.

However, during each freeze–thaw cycle, the free SH content of the samples fortified with lactylated gluten was significantly lower than that of the blank group, indicating that the addition of lactylated gluten could significantly reduce the release of free SH and maintain a high level of disulfide bond content. This might be due to the addition of lactylated gluten, which significantly reduced the content of freezable water in the dough. It reduced the degree of damage to the gluten network. This was consistent with the above analysis of the change of freezable water content in frozen dough. Meanwhile, compared with fresh dough, the free SH increased by 55.85% after three freeze–thaw cycles in the blank dough, while the free SH increased by 43.57% in the dough with sodium lactate-modified protein. It indicated that the addition of sodium lactate-modified gluten could slow down the trend of free SH increase caused by freeze–thaw cycles. However, in Lu et al. ’s [6] study, after 4 F/T, frozen dough with PUTY had even less content of SH than fresh dough. In summary, the implication of these results is that the lactylated gluten did not work as well as PUTY. However, to a certain extent, it could stabilize the disulfide bond in the dough during the freeze–thaw cycle.

### 3.6. Protein Network Microstructure

CLSM is a suitable method to observe the changes in the microstructure of the dough gluten network during freeze–thaw cycles [32]. Figure 7A shows the microscopic structure of frozen dough observed by CLSM during the freeze-thaw process. The red part of the figure represents the three-dimensional network structure of gluten. The black part represents the gap formed by gluten fracture during the freeze–thaw process, and some parts were undyed starch molecules. As shown in Figure 7A, the gluten network structure of fresh dough was relatively complete and orderly. However, with the increase in the number of freeze-thaw cycles, the gluten network broke to form many individual fragments. This might be due to the redistribution of water in the dough during the freeze-thaw cycle. The increase in the proportion of free water led to the increase in the recrystallization of ice, which caused certain damage to the gluten protein network [33]. This was consistent with the above results of changes in water distribution and frozen water content. The complete and continuous gluten network contributed to improving the gas retention of the dough so that the final product had a higher specific volume, while the temperature fluctuation during transportation affected the quality of steamed bread.

According to the CLSM image, after three freeze-thaw cycles, the gluten network microstructure of the dough to which lactylated gluten was added was still relatively compact and orderly compared with that of the blank group and the dough to which wheat gluten was added. This might be due to the addition of lactylated gluten protein, which could reduce the increase of free SH caused by freeze-thaw cycles. The frozen dough maintained a high degree of disulfide crosslinking, which was used to maintain the integrity of the gluten network structure. At the same time, because lactylated gluten reduced the number of large ice crystals formed in the frozen dough storage process, ice crystals’ destructive ability was reduced to a certain extent.

PNA can overcome the difficulty of quantifying CLSM images. Based on Figure 7B and Table 4, the quantitative analysis of the gluten network microstructure obtained similar results as above. After freeze-thaw cycles, the protective effect of lactylated gluten was better than that of wheat gluten. Specifically, after three freeze-thaw cycles, the total area and the number of protein network connection points of the dough adding lactylated gluten were significantly greater than those of the blank group and the WG group (*p* < 0.05). According to Lu et al. [4], after 4 F/T, frozen dough with NaHCO_3_ had 50% more protein network connection points than the blank group. However, in this study, frozen dough with lactylated had 24% more protein network connection points than the blank group. To sum up, the complete and continuous three-dimensional net structure plays a crucial part in the quality of the final steamed bread [34]. To some degree, the addition of lactylated gluten contributed to maintaining the dense microstructure of the steamed bread and thus improved the quality of the final product.

## 4. Conclusions

The influence of lactylated gluten and Freeze-Thaw Cycles on the microscopic properties of frozen dough and the texture properties of the corresponding product was investigated. The Freeze-Thaw Cycles caused the quality of steamed bread to decline. However, the freeze-thaw tolerance of frozen dough was improved by the addition of lactylated gluten. Specifically, the steamed bread made from frozen dough with the addition of lactylated gluten possessed a larger specific volume and a better texture property. Moreover, lactylated gluten could slow down the downtrend of the water–solid interaction caused by freeze–thawing, thus reducing water mobility. With the further addition of lactylated gluten, the interaction between protein and protein enhanced, resulting in forming the denser gluten protein network structure after Freeze-Thaw Cycles. In conclusion, frozen dough with lactylated gluten had better endurance in freeze–thaw conditions.

## Figures and Tables

**Figure 1 foods-12-03607-f001:**
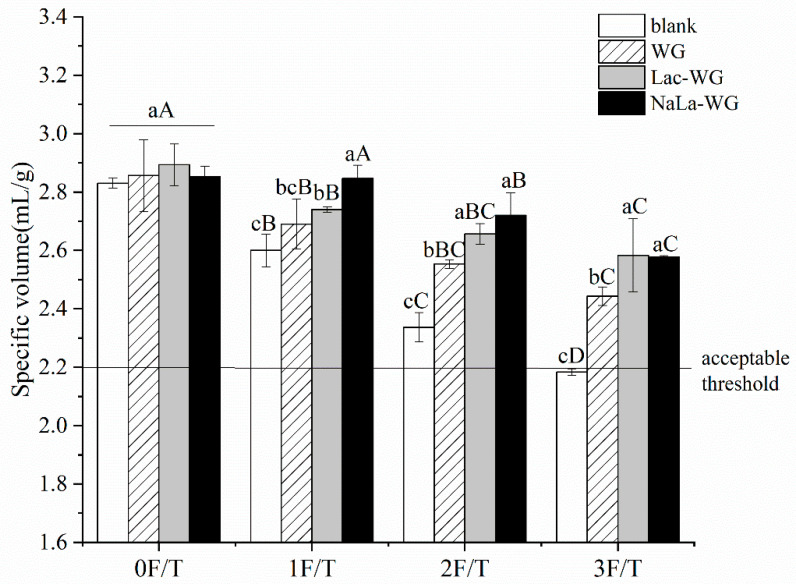
Effects of lactylated wheat gluten on the specific volume of steamed bread after freeze-thaw cycles. The bars indicate the standard error. The difference between the number of freeze-thaw cycles is indicated by capital letters, and the difference between different processing methods is indicated by lowercase letters.

**Figure 2 foods-12-03607-f002:**
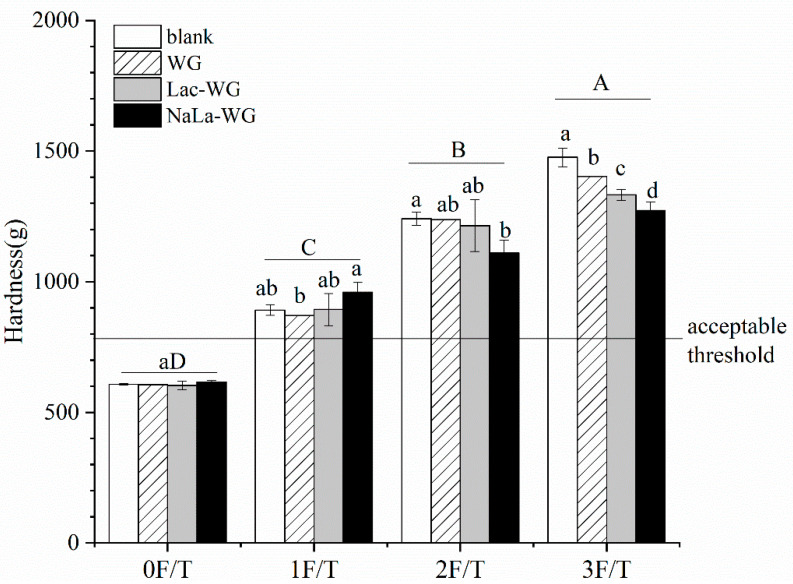
Effects of lactylated wheat gluten on the hardness of steamed bread after freeze-thaw cycles. The difference between the number of freeze-thaw cycles is indicated by capital letters; the difference between different processing methods is indicated by lowercase letters.

**Figure 3 foods-12-03607-f003:**
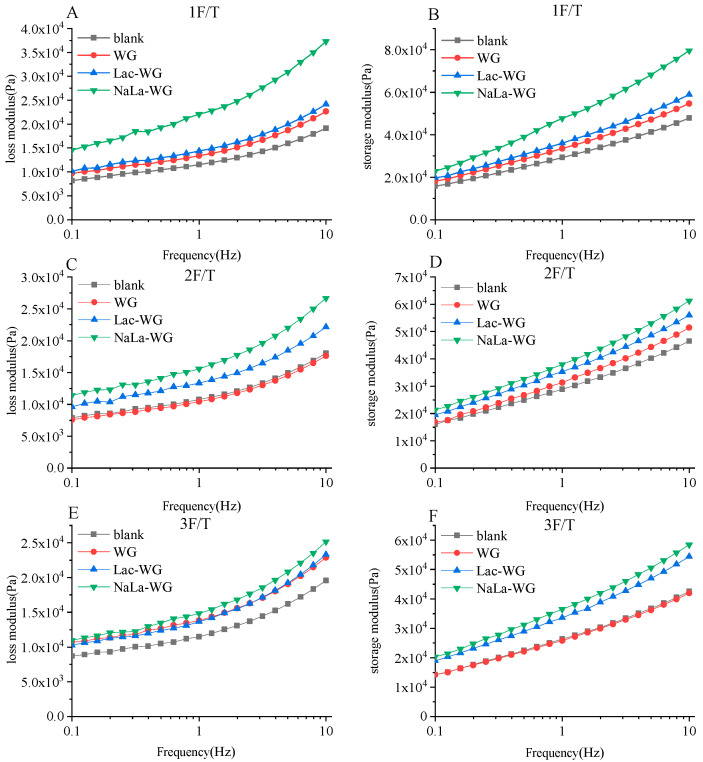
Effects of lactylated wheat gluten on dynamic rheological properties of frozen dough after freeze–thaw cycles. Loss modulus (**A**,**C**,**E**) and storage modulus (**B**,**D**,**F**) of 1 F/T (**A**,**B**), 2 F/T (**C**,**D**) and 3 F/T (**E**,**F**) frozen dough with different processing methods.

**Figure 4 foods-12-03607-f004:**
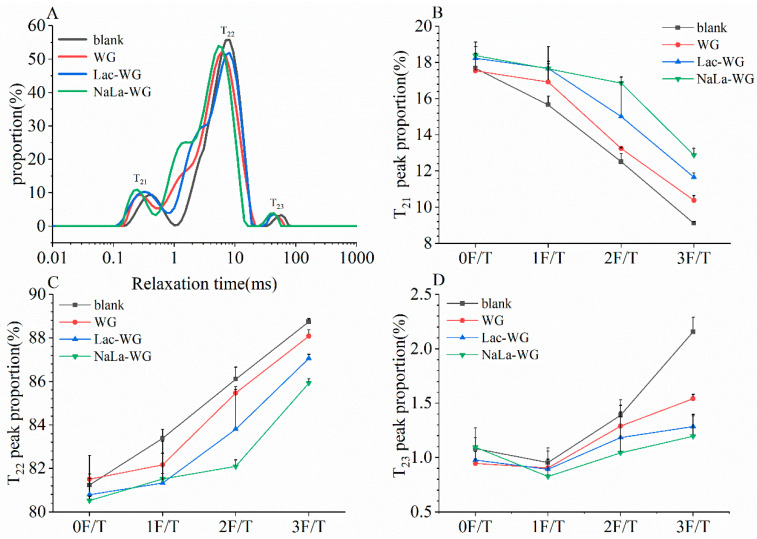
Effects of lactylated wheat gluten on the moisture distribution of frozen dough after freeze-thaw cycles. (**A**) Hydrogen proton group curves of frozen dough with different gluten added in the same freeze-thaw cycle determined by LF-NMR. (**B**) The proportion of strongly bound water. (**C**) The proportion of weakly bound water. (**D**) The proportion of free water.

**Figure 5 foods-12-03607-f005:**
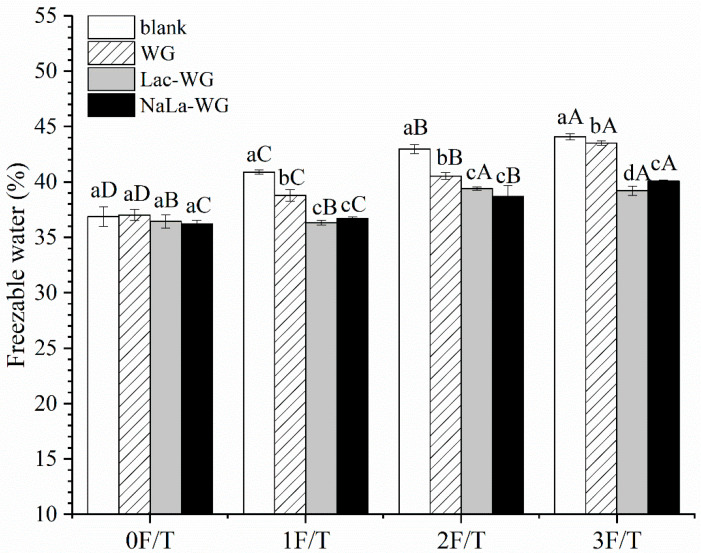
Effects of lactylated wheat gluten on the freezable water of frozen dough after freeze-thaw cycles. The difference between the number of freeze-thaw cycles is indicated by capital letters; the difference between different processing methods is indicated by lowercase letters.

**Figure 6 foods-12-03607-f006:**
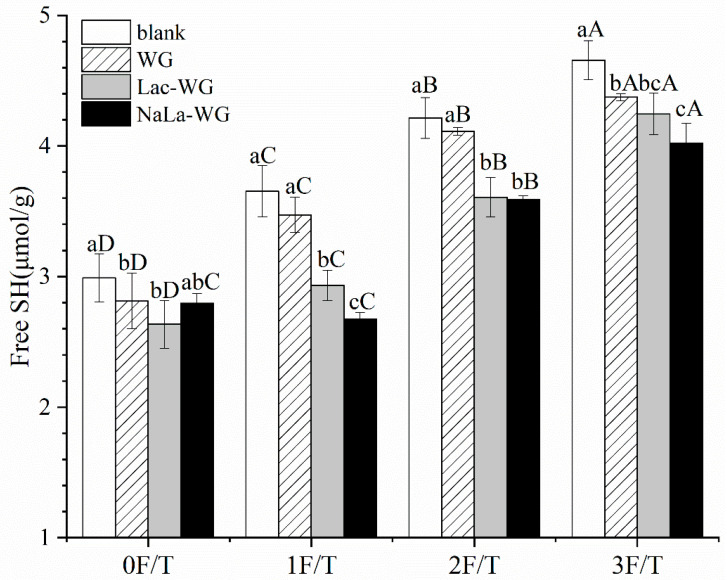
Effects of lactylated wheat gluten on free SH of frozen dough after freeze-thaw cycles. The difference between the number of freeze-thaw cycles is indicated by capital letters; the difference between different processing methods is indicated by lowercase letters.

**Figure 7 foods-12-03607-f007:**
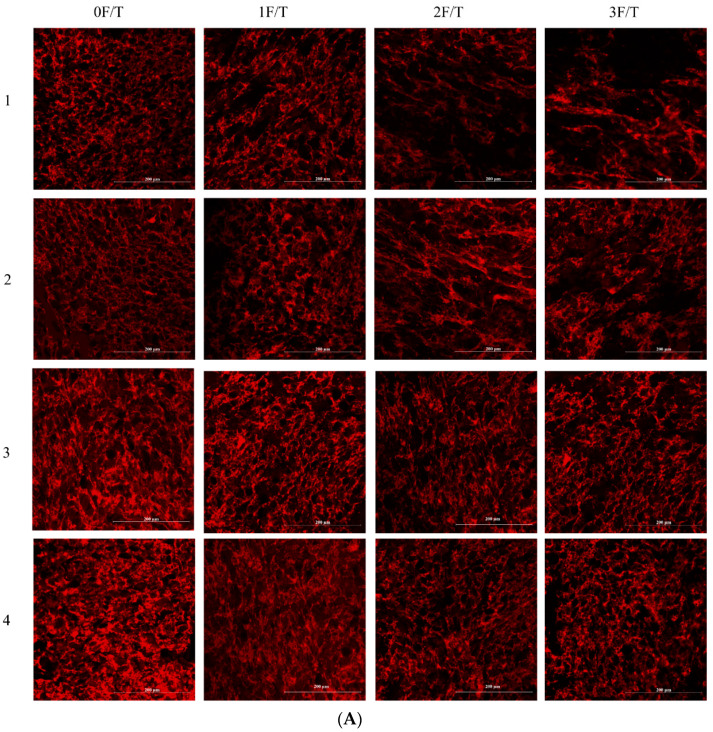
(**A**) Effects of lactylated wheat gluten on the original CLSM image of frozen dough after freeze-thaw cycles (1 is the blank group, 2 is the dough with untreated gluten added, 3 is the dough with lactate-treated gluten added, and 4 is the dough with sodium lactate-treated gluten added). (**B**) PNA analysis image.

**Table 1 foods-12-03607-t001:** Effects of lactylated wheat gluten on the springiness of steamed bread after freeze–thaw cycles.

	Blank	WG	Lac-WG	NaLa-WG
0F/T	0.94 ± 0.00 ^aA^	0.95 ± 0.01 ^aA^	0.98 ± 0.00 ^bA^	0.98 ± 0.01 ^bA^
1F/T	0.92 ± 0.00 ^cB^	0.94 ± 0.00 ^bA^	0.94 ± 0.01 ^bB^	0.97 ± 0.01 ^aA^
2F/T	0.90 ± 0.00 ^dC^	0.91 ± 0.00 ^cB^	0.92 ± 0.00 ^bC^	0.95 ± 0.00 ^aB^
3F/T	0.86 ± 0.00 ^cD^	0.88 ± 0.00 ^bcC^	0.89 ± 0.01 ^abD^	0.90 ± 0.00 ^aC^

The difference between the number of freeze-thaw cycles is indicated by capital letters; the difference between different processing methods is indicated by lowercase letters.

**Table 2 foods-12-03607-t002:** Effects of lactylated wheat gluten on the surface color of steamed bread after freeze-thaw cycles.

	Group	L*	a*	b*
0F/T	blank	85.82 ± 1.06 ^aA^	−0.17 ± 0.02 ^aA^	19.32 ± 0.60 ^aB^
	WG	85.56 ± 0.63 ^aA^	−0.17 ± 0.01 ^aB^	19.37 ± 0.70 ^aB^
	Lac-WG	86.93 ± 0.91 ^aA^	−0.16 ± 0.03 ^aA^	19.06 ± 0.62 ^aB^
	NaLa-WG	85.82 ± 1.00 ^aA^	−0.14 ± 0.02 ^aA^	20.33 ± 0.96 ^aA^
1F/T	blank	84.73 ± 0.18 ^abB^	−0.16 ± 0.03 ^aA^	19.96 ± 0.51 ^aAB^
	WG	84.48 ± 0.40 ^aB^	−0.17 ± 0.00 ^aB^	18.39 ± 0.15 ^bC^
	Lac-WG	84.25 ± 0.24 ^aB^	−0.16 ± 0.05 ^aA^	18.14 ± 0.14 ^bB^
	NaLa-WG	84.68 ± 0.69 ^aAB^	−0.17 ± 0.02 ^aA^	18.45 ± 0.22 ^bAB^
2F/T	blank	83.64 ± 0.21 ^aC^	−0.16 ± 0.03 ^aA^	20.40 ± 0.15 ^aAB^
	WG	82.80 ± 0.74 ^aC^	−0.13 ± 0.03 ^aA^	20.20 ± 0.18 ^aAB^
	Lac-WG	83.55 ± 0.25 ^aB^	−0.18 ± 0.03 ^aA^	20.06 ± 0.42 ^aA^
	NaLa-WG	83.56 ± 0.51 ^aBC^	−0.14 ± 0.01 ^aA^	19.70 ± 1.04 ^aAB^
3F/T	blank	81.59 ± 0.07 ^cD^	−0.15 ± 0.00 ^aA^	20.99 ± 0.96 ^aA^
	WG	82.44 ± 0.21 ^bC^	−0.17 ± 0.02 ^aA^	20.83 ± 0.60 ^aA^
	Lac-WG	82.28 ± 0.07 ^bC^	−0.16 ± 0.01 ^aAB^	20.31 ± 0.72 ^aA^
	NaLa-WG	83.22 ± 0.43 ^aC^	−0.15 ± 0.04 ^aA^	18.85 ± 0.18 ^bAB^

The difference between the number of freeze-thaw cycles is indicated by capital letters; the difference between different processing methods is indicated by lowercase letters.

**Table 3 foods-12-03607-t003:** Effects of lactylated wheat gluten on the freezable water of frozen dough after freeze-thaw cycles.

	Group	∆H	Wt/%	Fw/%
0F/T	blank	50.58 ± 0.38 ^aC^	41.12 ± 1.10 ^aA^	36.88 ± 0.87 ^aD^
	WG	50.81 ± 0.05 ^aD^	41.22 ± 0.52 ^aA^	37.02 ± 0.50 ^aD^
	Lac-WG	50.00 ± 0.09 ^bB^	41.21 ± 0.62 ^aA^	36.44 ± 0.60 ^aB^
	NaLa-WG	50.00 ± 0.08 ^bB^	41.44 ± 0.40 ^aA^	36.23 ± 0.31 ^aC^
1F/T	blank	55.26 ± 0.31 ^aB^	40.59 ± 0.40 ^aA^	40.89 ± 0.19 ^aC^
	WG	52.65 ± 0.12 ^bC^	40.77 ± 0.53 ^aA^	38.79 ± 0.53 ^bC^
	Lac-WG	48.99 ± 0.28 ^cC^	40.51 ± 0.45 ^aA^	36.32 ± 0.21 ^cB^
	NaLa-WG	49.73 ± 0.27 ^dB^	40.70 ± 0.04 ^aA^	36.70 ± 0.16 ^cC^
2F/T	blank	55.77 ± 0.28 ^aB^	38.97 ± 0.51 ^aB^	42.98 ± 0.41 ^aB^
	WG	53.22 ± 0.08 ^bB^	39.43 ± 0.32 ^aB^	40.53 ± 0.31 ^bB^
	Lac-WG	51.64 ± 0.27 ^cA^	39.35 ± 0.07 ^aB^	39.41 ± 0.13 ^cA^
	NaLa-WG	50.99 ± 1.33 ^cB^	39.56 ± 0.26 ^aC^	38.71 ± 0.95 ^cB^
3F/T	blank	57.23 ± 0.45 ^aA^	38.99 ± 0.10 ^bB^	44.08 ± 0.26 ^aA^
	WG	56.54 ± 0.26 ^bA^	39.01 ± 0.03 ^bB^	43.52 ± 0.20 ^bA^
	Lac-WG	51.34 ± 0.42 ^dA^	39.30 ± 0.14 ^aB^	39.22 ± 0.43 ^dA^
	NaLa-WG	52.47 ± 0.15 ^cA^	39.30 ± 0.08 ^aC^	40.10 ± 0.07 ^cA^

The difference between the number of freeze-thaw cycles is indicated by capital letters; the difference between different processing methods is indicated by lowercase letters.

**Table 4 foods-12-03607-t004:** The PNA results of lactylated wheat gluten on frozen dough.

	Group	Total Protein Network Area (%)	Protein Network Connection Points
0F/T	blank	36.51 ± 0.11 ^cA^	677 ± 16 ^cA^
	WG	37.19 ± 0.08 ^bA^	722 ± 22 ^bA^
	Lac-WG	37.84 ± 0.57 ^bA^	781 ± 3 ^bA^
	NaLa-WG	39.66 ± 0.40 ^aA^	821 ± 27 ^aA^
1F/T	blank	30.88 ± 0.06 ^dB^	644 ± 20 ^cB^
	WG	33.67 ± 0.25 ^cB^	669 ± 13 ^bcB^
	Lac-WG	35.47 ± 0.12 ^bB^	690 ± 15 ^bB^
	NaLa-WG	38.14 ± 0.05 ^aB^	738 ± 26 ^aB^
2F/T	blank	22.72 ± 0.34 ^dC^	589 ± 10 ^cC^
	WG	31.36 ± 0.43 ^cC^	610 ± 9 ^bC^
	Lac-WG	33.69 ± 0.08 ^bC^	634 ± 11 ^aC^
	NaLa-WG	35.40 ± 0.03 ^aC^	652 ± 8 ^aC^
3F/T	blank	22.58 ± 0.19 ^dC^	481 ± 14 ^cD^
	WG	28.39 ± 0.37 ^cD^	529 ± 17 ^bD^
	Lac-WG	32.50 ± 0.38 ^bD^	577 ± 13 ^aD^
	NaLa-WG	35.21 ± 0.03 ^aC^	598 ± 5 ^aD^

The difference between the number of freeze-thaw cycles is indicated by capital letters; the difference between different processing methods is indicated by lowercase letters.

## Data Availability

The data used to support the findings of this study can be made available by the corresponding author upon request.

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
