# Peer review of "Effect of Lactylated Gluten and Freeze-Thaw Cycles on Frozen Dough: From Water State and Microstructure"

_foods, 2023, doi:10.3390/foods12193607_

Round 1
Reviewer 1 Report
In the current manuscript, the effect of lactylated gluten and freeze–thaw cycles on the water state, microstructure and quality of frozen steamed bread dough was studied. The article has enough experiments and the results are clearly presented. In my opinion, although The materials and methods section needs further clarification, with some minor corrections.
My comments are as follows:
Abstract:
- P1-L7: What is SH? The full name must be written for the first time.
- In one sentence and at the end of the abstract, the conclusion or recommendation of this research should be written.
- Use keywords that are not in the title of the article.
Introduction:
- Add a reference at the end of the first paragraph.
- The sentences written in the second paragraph need more references.
- P2-last paragraph of Introduction: After the specific volume, a number of references should be mentioned.
- P2-last paragraph of Introduction: Therefore, the aim of this study … .
Materials & Methods
- The characteristics of the yeast should be written. Genus and species are important.
- What is " frozen green dough" ? more explain.
- How was it determined that 0.5 % of the wheat gluten should be added?
- How much lacylate was added to the dough? Explained in the dough preparation section.
- Write the specifications of the refrigerator or freezer.
- How did you provide and adjust the temperature of 30°C for thawing?
- Write the specifications of the vacuum freeze dryer.
- After packing, what conditions were the samples kept in until the next tests?
- Write the specifications of the equipment used for steaming process.
- Add a reference for the specific volume of steamed bread procedure.
- After steaming, how and to what temperature was the bread cooled?
- Write the specific of the texture analyzer.
- Add a reference for measurement of free SH content of frozen dough section.
- 2.3.3. Please add more details.
Results & Discussion
- In this section, the results of each test should be compared with the observations of other researchers.
Author Response
In the current manuscript, the effect of lactylated gluten and freeze–thaw cycles on the water state, microstructure and quality of frozen steamed bread dough was studied. The article has enough experiments and the results are clearly presented. In my opinion, although The materials and methods section needs further clarification, with some minor corrections.
My comments are as follows:
Abstract:
P1-L7: What is SH? The full name must be written for the first time.
Reply: Thanks for reviewer’s reminder. We have marked the full name.
In one sentence and at the end of the abstract, the conclusion or recommendation of this research should be written.
Reply: Thanks for reviewer’s reminder. We have added the conclusion in revised manuscript.
Use keywords that are not in the title of the article.
Reply: Thanks for reviewer’s reminder. We have rewritten the keywords in revised manuscript.
Introduction:
Add a reference at the end of the first paragraph.
Reply: Thanks for reviewer’s reminder. We have added a reference in revised manuscript.
The sentences written in the second paragraph need more references.
Reply: Thanks for reviewer’s reminder. We have added the relevant references in revised manuscript.
P2-last paragraph of Introduction: After the specific volume, a number of references should be mentioned.
Reply: Thanks for reviewer’s reminder. We have added the relevant references in revised manuscript.
P2-last paragraph of Introduction: Therefore, the aim of this study … .
Reply: Thanks for reviewer’s reminder. We have restructured this sentence in revised manuscript.
Materials & Methods
The characteristics of the yeast should be written. Genus and species are important.
Reply:Thanks for reviewer’s reminder. The details of the yeast were added in revised manuscript.
What is " frozen green dough" ? more explain.
Reply: Frozen dough refers to semi-finished products formed by kneading and quick-freezing, and must be reprocessed in the later stage. Analysing the index of frozen dough can better understand the protective effect of protein on dough.
How was it determined that 0.5 % of the wheat gluten should be added?
Reply: In other references, the amount of antifreeze protein added is usually in the range of 0.1%-1%, so 0.5 % of the wheat gluten were added.
How much lacylate was added to the dough? Explained in the dough preparation section.
Reply: The lactylated wheat gluten was prepared according to Wang et al. This was already stated in the 2.1.
Write the specifications of the refrigerator or freezer.
Reply:Thanks for reviewer’s reminder. The details of the refrigerator were added in revised manuscript.
How did you provide and adjust the temperature of 30°C for thawing?
Reply:Using the fermentation room to adjust the temperature. The details were added in revised manuscript.
Write the specifications of the vacuum freeze dryer.
Reply:Thanks for reviewer’s reminder. The details of the vacuum freeze dryer were added in revised manuscript.
After packing, what conditions were the samples kept in until the next tests?
Reply:Freeze-dried samples werestored in a ziplock bag in a dryer. Frozen dough were stored in -18°C refrigerator until the next tests.
Write the specifications of the equipment used for steaming process.
Reply:Thanks for reviewer’s reminder. The specifications of the equipment used for steaming process were added in revised manuscript.
Add a reference for the specific volume of steamed bread procedure.
Reply:Thanks for reviewer’s reminder. The reference was added in revised manuscript.
After steaming, how and to what temperature was the bread cooled?
Reply:The sample was cooled for 1 h at room temperature before analysis. The details were added in revised manuscript.
Write the specific of the texture analyzer.
Reply:Thanks for reviewer’s reminder. The details of the texture analyzer were added in revised manuscript.
Add a reference for measurement of free SH content of frozen dough section.
Reply:Thanks for reviewer’s reminder. The details of the measurement of free SH content were added in revised manuscript.
2.3.3. Please add more details.
Reply:Thanks for reviewer’s reminder. The details were added in revised manuscript.
Results & Discussion
In this section, the results of each test should be compared with the observations of other researchers.
Reply:Thanks for reviewer’s reminder. The discussions were added in revised manuscript.
Reviewer 2 Report
In my opinion, the presented manuscript still needs improvement.
The issues presented in it related to the modification of gluten before freezing the dough are interesting and may contribute to a significant contribution to the development of this technology. There are, however, a few issues that need to be addressed.
Firstly, the introduction is not a contribution to the conducted research. Although the authors devoted a lot of space to the issue of strengthening gluten, they nevertheless omitted issues related to lactylated gluten, however, as we learn from the purpose, they had already conducted preliminary research.
Paragraph 2.3.2 mentions the texture analyzes performed. There is no information about the type and model of the device. Has a typical TPA profile been run?
Similarly, in paragraph 2.3.3 there is no information about the details of the color measurement. Observer?, Illuminant?
In the part with the results, I think that the color results were presented incorrectly, the delta is missing, the value of which is the only basis for inferring significant changes in the CIELab scale.
As for the texture measurement, I also think that the results are not properly presented. In the graph, we only have results for hardness. Was only this parameter measured. Elasticity is also mentioned in the text
Author Response
In my opinion, the presented manuscript still needs improvement. The issues presented in it related to the modification of gluten before freezing the dough are interesting and may contribute to a significant contribution to the development of this technology. There are, however, a few issues that need to be addressed.
Firstly, the introduction is not a contribution to the conducted research. Although the authors devoted a lot of space to the issue of strengthening gluten, they nevertheless omitted issues related to lactylated gluten, however, as we learn from the purpose, they had already conducted preliminary research.
Reply:Thanks for reviewer’s reminder. The details of lactylated gluten were added in revised manuscript.
Paragraph 2.3.2 mentions the texture analyzes performed. There is no information about the type and model of the device. Has a typical TPA profile been run?
Reply:Thanks for reviewer’s reminder. The information of texture analyzes were added in revised manuscript.
Similarly, in paragraph 2.3.3 there is no information about the details of the color measurement. Observer?, Illuminant?
Reply:Thanks for reviewer’s reminder. The information of color measurement were added in revised manuscript.
In the part with the results, I think that the color results were presented incorrectly, the delta is missing, the value of which is the only basis for inferring significant changes in the CIELab scale.
Reply:Thanks for reviewer’suggestion. There are so many studies that use this data to characterize color changes.
He Y J, Guo J Y, Ren G Y, et al. Effects of konjac glucomannan on the water distribution of frozen dough and corresponding steamed bread quality[J]. Food Chemistry, 2020, 330: 127243.
As for the texture measurement, I also think that the results are not properly presented. In the graph, we only have results for hardness. Was only this parameter measured. Elasticity is also mentioned in the text.
Reply:Thanks for reviewer’suggestion. Hardness is the force exerted on food when the teeth chew it and is considered to be a key factor in consumer acceptability of a product. In addition, the biggest effect of freeze-thawing on dough is the increase of Hardness. So, we focused on changes in hardness.
Carocho M, Morales P, Ciudad-Mulero M, et al. Comparison of different bread types: Chemical and physical parameters[J]. Food Chemistry, 2020, 310: 125954.
Lu L, Yang Z, Guo X N, et al. Effect of NaHCO3 and freeze–thaw cycles on frozen dough: From water state, gluten polymerization and microstructure[J]. Food Chemistry, 2021, 358: 129869.
Round 2
Reviewer 2 Report
Most of my comments on the original version of the manuscript have been taken into account by the authors. As for other issues, it accepts the authors' explanations. I recommend the manuscript for publication